# Separation of Cenospheres from Lignite Fly Ash Using Acetone–Water Mixture

**Sorachon Yoriya \*** , **Tanasan Intana and Phattarathicha Tepsri**

National Metal and Materials Technology Center, 114 Thailand Science Park, Pahonyothin Road, Khlong Neuang, Khlong Luang, Pathum Thani 12120, Thailand
\* Correspondence: sorachy@mtec.or.th; Tel.: +66-2564-6500 (ext. 4224)

**Abstract:** This work reports on the separation of cenospheres from lignite fly ash through a wet separation process-the sink-float method. A better quality of cenospheres could be achieved through a physical–chemical approach using an acetone–water mixture as a medium. This work aimed to elucidate the correlation between the structure, morphology, and composition and medium fraction variables, with data for the freshly prepared and the reused mixtures presented for comparison. The work covers a study of the macrocomponent composition of an $Fe_2O_3$–$SiO_2$–$Al_2O_3$ system, highlighting the pair dependences of $SiO_2$–$Al_2O_3$, $Al_2O_3$–$Fe_2O_3$, and $SiO_2/Al_2O_3$–$Al_2O_3$ and revealing an interesting result in terms of geochemical characteristics categorizing the collected cenosphere fractions separated from high-calcium class C fly ash produced from a lignite coal power plant in Thailand (as magnetic cenospheres). The CaO and $SO_3$ contents increased monotonically with increased water content, particularly for the CaO composition profile, which was found to be similar to the increased carbonate concentration measured from the mixtures after use. The physicochemical properties in terms of the self-association ability of the acetone–water mixing phase is believed to have played an important role in determining the intermolecular interactions and reactivity of ions in the liquid phase, consequently affecting the separation efficiency, recovery yield, and quality of cenospheres.

**Keywords:** cenospheres; separation; fly ash; $CaCO_3$; acetone–water; chemical composition; morphology

## 1. Introduction

Fly ash, a byproduct from coal-fired thermal power plants, is a pozzolanic material recognized as a valuable resource that can be added to construction materials to form cementitious compounds when in contact with water [1–3]. Cenospheres, hollow, spherically shaped particles that are mostly open-pore type in nature, are one of the most important value-added materials or subproducts that mix with fly ash. This is due to the distinctive properties of cenospheres, such as being lightweight, good flowability, chemically inertness, good insulation, high compressive strength, and low thermal conductivity, which enable them to be widely used in many industrial applications. Cenospheres have become a highly in demand material as a filler or additive in many specialized applications, such as in lightweight cements [4,5], polymeric composites [6–10], automotive brake rotors and differential covers [11], mullite-coated diesel engine components [12,13], and electromagnetic shielding and energy absorption applications [14,15]. Applications for cenospheres as a construction material have been found, such as in lightweight thermal insulation composite [16], lightweight sound-absorbing structural material with cenosphere-reinforced cement and asphalt concrete [17], and lightweight concrete [18,19]. It has been recently reported that cenospheres are used as an additive or filler in polymer concrete matrix for manufacturing composite beams [20] and composite railway sleepers [21].

With their spherical and hollow morphologies, cenospheres are particularly promising, with high resistance to crack propagation.

The density of cenospheres varies from 0.2 g/cc up to 2.6 g/cc [22–24]. The availability of such low-density (<1 g/cc) cenospheres is quite low, as they constitute a small fraction of about 0.3%–1.5% by weight in fly ash [25–30]. A higher density up to 2.9 g/cc could possibly be found for the cenosphere type containing heavy iron oxides in the silica matrix of the spheres [28]. The particle size of cenospheres is much larger than fly ash particles and can range from 5 μm to 500 μm in diameter. The shell thickness varies from 2 μm to 30 μm [27,31–34]. Shape, size, and density are the three major characteristics putting cenospheres in high demand. Cenospheres are of a similar chemical composition to fly ash, with their mineralogical composition depending on the geological features of the coal source and the reactions occurring as a result of combustion conditions that determine how the minerals fuse and form the solid hollow aluminosilicate spheres. The main minerals in fly ash are quartz, mullite, hematite, magnetite, and calcite, while the crystalline phases consist of gypsum, aluminum oxides, chlorite, feldspars, iron-bearing oxides, spinel ($FeAl_2O_4$), and mullite [3,34]. Cenospheres formed with a high content of mullite could be used to form cementitious compounds to enhance durability in concrete work [12].

The existing methods of cenosphere separation are classified into two groups: wet separation and dry separation. The wet separation mechanism is based on the differences between the density of solid particles and a liquid medium: with this process, cenospheres can be recovered from fly ash through a gravity settling separation-the sink-float method. The separation efficiency of wet separation is based on the natural buoyancy of cenospheres in the medium, the concentration of feeding particles, the surface topology and porosity of particles, and refinement cycles [30]. However, the efficiency is limited by the influence of the dense mass of fly ash, which prevents particles lighter than water from escaping the agglomeration, consequently hindering the rise of the lighter particles to the surface. The advantage of the wet separation method is the capability of obtaining low-density, intact cenospheres directly from the separation process: nonetheless, the availability of land and water is a major concern for large-scale production. Another problem is the issue of dissolution of hazardous materials into water sources [35,36]. A further drawback is the need for an additional drying step before further use. In terms of the material quality (particularly for class C fly ash with a high calcium content), crystals are formed on the particle surface and then become hardened in the drying stage, consequently limiting their use for further applications. The dry separation is an alternative method aiming to overcome the problems of the wet separation. With this method, the chemical composition remains unchanged. In addition, there is no drying stage needed, thus avoiding energy consumption issue: nonetheless, this method requires advanced technology involving pneumatic separation, such as an air classifier, to efficiently classify the particles [37]. Air classification is an operation that separates dispersed solid particles based on their differences in size, geometrical shape, and density in the air stream.

Focusing herein on the traditional and facile process, in the wet separation, water is a widely used medium to collect lightweight cenosphere particles from fly ash due to simplicity and convenient availability [29]. Cenospheres are glass spheres with a thin wall and are generally regarded as a lighter material than water with a relative density of less than 1.0 g/cc: thus, they float on the water surface and can be separated by skimming off the floating layer. Cenospheres of different densities lighter and heavier than water have been found to vary in their physico–chemico–mineralogical properties [18]. Tailoring appropriate density ranges of the medium specific to the large fraction of cenospheres to be collected has come into focus. Different liquid densities have been examined to enhance the recovery percentage in terms of the mass fractions of cenospheres collected from fly ash. Different ranges of density of the prepared mixtures have been reported in the literature. These include acetone (0.789 g/cc)–water and acetone-triethanol amine lauryl sulfate (TEALS) [23], water–zinc chloride (1.10–1.80 g/cc), lithium metatungstate (LMT) (1.5 g/cc)–water [24], a carbon tetrachloride-dibromomethane mixture (2.2 g/cc), dibromomethane (2.5 g/cc), and di-iodomethane

(3.3 g/cc) [33]. To some extent, single-liquid flotation seems to be more likely not to give efficient separation of all cenosphere fractions from total fly ash. Two types of liquids with difference densities are normally adopted specifically to meet particular purposes, such as obtaining controllable particle size ranges and narrow-range density. For instance, in a water–LMT mixture, water was employed for separating the floaters of density of less than 1.0 g/cc, while LMT was specifically used for separating the floaters of density of greater than 1.0 g/cc but less than 1.5 g/cc [24]. Dilution of the solution could be prepared by mixing heavy liquids in different proportions: heavy cenospheres with a density range between 1.0 and 3.0 g/cc could be obtained [24].

Using water as a medium, cenosphere separation from low-calcium class F fly ash is mostly seen in the literature [22,24,27]. However, for class C fly ash, the use of water for cenosphere separation appears to have become of disadvantageous concern: when mixed with water, the soaked layer rapidly hardens and cannot be used further for other purposes [24]. The high calcium composition of class C fly ash inevitably leads to the formation of crystals, with a significant amount on the particle surface. Thus, special attention has been given to the use of nonaqueous solvents as liquid medium with the purpose of inhibiting the crystalline formation, which deteriorates the quality and recovery yield of cenospheres. Understanding of the properties of cenospheres and their specific areas of application is necessary to consider their use as a relatively cheaper replacement material in manufacturing industries [32].

This work focused on the influence of an acetone–water mixture on cenosphere recovery from high-calcium class C fly ash containing calcium oxide >20 wt %. A main purpose of this work was oriented toward ascertaining the possibilities of improving the quality of cenospheres collected from high-calcium fly ash. The physical and chemical properties of cenospheres correlating with the properties of the medium and the medium fractions were also a non-negligible factor in understanding their effects on the separation efficiency that ultimately limited the cenosphere recovery yield. The relationship between morphology and composition will be described.

## 2. Materials and Methods

The fly ash used in this work was class C fly ash taken from a lignite coal-fired power plant in Mae Moh, Lampang, Thailand. The bulk density of the fly ash was 1.3 g/cc, and the apparent density was 2.48 g/cc. The chemical compositions of fly ash are given in Table 1. The lime content of the fly ash sample was 1.43%; this slight amount can result in high alkaline solutions.

**Table 1.** Calculated density of acetone–water mixtures at different ratios of acetone and water, the percentage of cenosphere recovery of cenospheres collected from using different ratios of freshly prepared and used acetone–water mixtures (4-h soaking period), and fly ash/medium ratios of 1:2.5, 1:5, and 1:10.

| Acetone/Water Ratio (% *v/v*) | Calculated Density of Acetone–Water Mixtures | Percentage of Cenosphere Recovery (%) | | | | | |
|---|---|---|---|---|---|---|---|
| | | Fly Ash/Medium Ratio | | | | | |
| | | 1:2.5 (Freshly Prepared) | 1:2.5 (Used) | 1:5 (Freshly Prepared) | 1:5 (Used) | 1:10 (Freshly Prepared) | 1:10 (Used) |
| 100:0 | 0.79 | 0.01 ± 0.001 | 0.01 ± 0.001 | 0.03 ± 0.004 | 0.02 ± 0.002 | 0.02 ± 0.003 | 0.03 ± 0.003 |
| 80:20 | 0.85 | 0.02 ± 0.001 | 0.04 ± 0.001 | 0.06 ± 0.002 | 0.04 ± 0.003 | 0.07 ± 0.003 | 0.04 ± 0.002 |
| 50:50 | 0.90 | 0.06 ± 0.003 | 0.03 ± 0.002 | 0.15 ± 0.020 | 0.10 ± 0.015 | 0.11 ± 0.008 | 0.12 ± 0.005 |
| 20:80 | 0.96 | 0.31 ± 0.005 | 0.15 ± 0.025 | 0.32 ± 0.008 | 0.29 ± 0.014 | 0.38 ± 0.007 | 0.31 ± 0.023 |

Prior to cenosphere separation, the fly ash sample was oven-dried at 105 °C for 24 h. The separation of cenospheres was performed through the wet separation process sink-float method. The medium used to separate the cenospheres from fly ash was a mixture of acetone and water. The acetone (RCI Labscan Limited, 99.5%)–water mixture was prepared by mixing in different proportions, and the liquid contents were varied in terms of the acetone/water ratio as follows: 100:0, 80:20: 50:50, 20:80, and 0:100% *v/v*. When water is mixed with acetone at different volume percentages, basically the density of the mixture decreases from 1.00 g/cc based upon the principle of an acetone–water mixture that behaves as a non-ideal solution [38]. The extent of the density decrease is directly dependent on the

volume percent of acetone added in the mixture. Accordingly, in this study, the density of the mixtures with different acetone contents mixed with water was calculated (see the values in Table 1).

The experiments were performed to determine the cenosphere recovery yield under the specified conditions: while the ratios of acetone and water were varied, the solid/liquid ratio was held fixed. The fly ash/medium ratios selected in this study were 1:2.5, 1:5, and 1:10 g/mL. Total volume per batch was 200 mL. While adding fly ash into the mixing container, manual stirring was applied immediately and continuously. The mixture was stirred for 10 min before leaving it for sedimentation, as the particles lighter than the medium density would float to the top surface and the heavier particles would sink. The recovery process to collect the floater part was done by stirring and settling repeatedly. Cenospheres were collected every two hours or two times during the four-hour period by carefully skimming off the floating layer into the receiving container. This collected part was drained through the vacuum filter and dried in the oven at 105 °C for 24 h. The percentage of cenosphere recovery was calculated by the following equation:

$$Percentage\ of\ cenospheres\ recovery\ (\%) = \frac{Weight\ of\ floating\ particles}{Weight\ of\ total\ fly\ ash} \times 100. \tag{1}$$

The collected cenospheres were characterized for their physical properties and chemical composition and through mineralogical and morphological studies. Particle size and distribution were determined by laser particle size analyzer (Malvern Mastersizer 2000). Density was examined with an ultrapycnometer (Ultrapycnometer 1200e) in a helium gas atmosphere. Chemical composition by means of oxide percentages of cenospheres was also determined by microenergy dispersive X-ray fluorescence (EDXRF) (EDAX model: ORBIS PC) using a 2-mm beam diameter for the bulk of the cenosphere sample and a 30-μm beam size on each particle (varying in sizes of 50, 100, 150, 200, 250, 300, 350, and 400 μm). Morphological structure was studied by scanning electron microscope (SEM), with elemental composition analyzed by energy dispersive X-ray spectroscopy (EDS) (Phenom ProX). The hollow structure of cenospheres was confirmed by SEM analysis. In addition, a carbonate alkalinity measurement was applied to this study in the evaluation of calcium carbonate solubility in various water contents, and the concentration of calcium carbonate was determined. The used mixtures were further quantitatively analyzed for carbonate alkalinity via the standard method, 2320 B, based on American Public Health Association (APHA), 2012, Standard methods for the examination of water and wastewater. A 500-mL portion of the medium samples was collected for the measurement.

## 3. Results and Discussion

### 3.1. Cenosphere Recovery and Density

A fraction of cenospheres was obtained from each range of density. The percentage of cenosphere recovery of all acetone/water ratios was determined and is shown in Table 1 and Figure 1a. The separation scheme under low-water conditions, 0%–50% *v/v*, which correlated with the relatively lower density of the mixtures, resulted in a lower recovery yield of cenospheres. With liquid densities <0.90 g/cc, the floaters were collected in a small fraction of less than 0.15%, whereas with liquid densities >0.90 g/cc, the cenosphere recovery significantly increased as water content in the medium increased. The dependence of recovery yield on water content was more pronounced when incorporating more water, from 50%–100% *v/v*, in the medium. Increasing water content up to 80% *v/v* in the mixture led to a high percentage about three times more than that of the 50% *v/v* content. At 100% water content, the percent recovery achieved was 0.36 wt %. In terms of the order of water variation, the increase in cenosphere recovery was quite significant. In addition, the reuse of the mixtures at the second time gave an insignificant result of percentage recovery compared to the results obtained from the freshly prepared mixtures, as seen in Figure 1a. A large deviation was clearly seen for the reused condition with a high concentration of the ash in the medium at 1:2.5: a relatively lower cenosphere recovery yield was obtained.

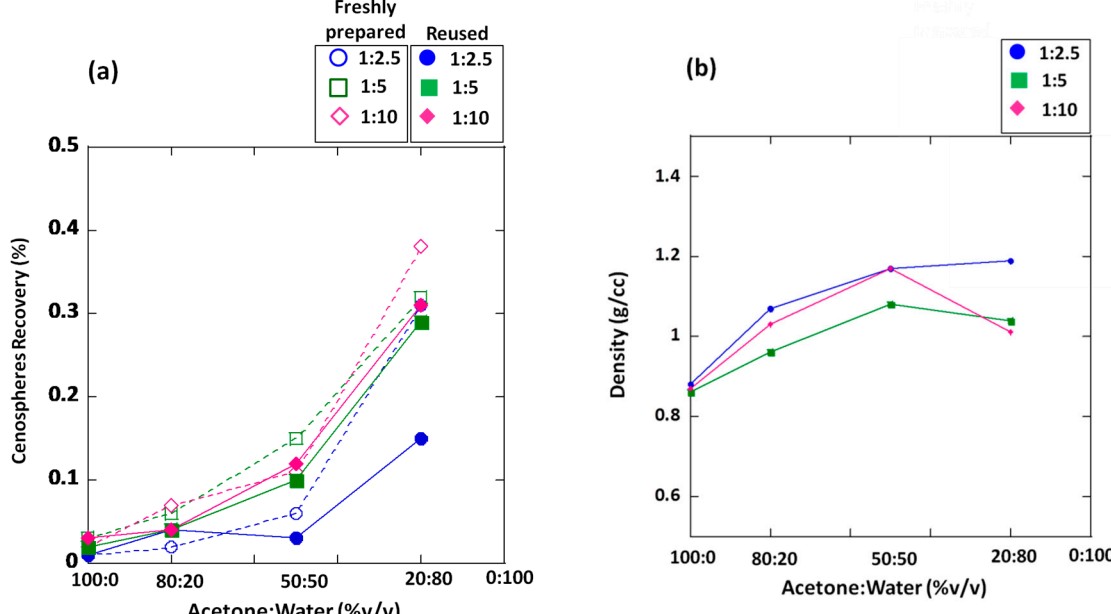

**Figure 1.** Plots of accumulative percentages of cenosphere recovery against (**a**) the ratio of the acetone–water mixture, comparing the results obtained from the freshly prepared and the reused mixtures, and (**b**) the density of the cenosphere fractions obtained from using the freshly prepared mixtures. All data points were obtained from five measurements, and the SD of all points shown in (**b**) is less than 0.005. Due to the small SD values demonstrated in Table 1 and in the results in Figure 1b, the error bars for the data in Figure 1a,b are fairly invisible. Cenospheres were collected from using different ratios of acetone–water mixtures, a 4-h soaking period, and fly ash/medium ratios of 1:2.5, 1:5, and 1:10.

Figure 1b shows the density of cenospheres collected from all ratios of the freshly prepared mixture. For the lighter cenospheres, which were obtained from the ratios of 100:0, 20:80, and 50:50 with their liquid density of <0.9 g/cc, the density was in the range of 0.8–1.2 g/cc. For the heavier cenospheres, obtained from the 20:80 ratio with its liquid density of >0.9 g/cc, the density was in a bit of a higher range, at 1.0–1.2 g/cc.

### 3.2. Particle Size and Distribution

Particle size and distribution were analyzed by laser particle size analyzer. The average size of particles was determined by means of the volume mean diameter, *D [4,3]*.

The particle size distribution of the fly ash used in this study widely varied (1–15 μm (38.40%), 15–30 μm (16.80%), 30–45 μm (10.10%), 45–60 μm (7.20%), 65–75 μm (5.10%), and >100 μm (22.40%)): the percentage value shown in parentheses conformed to the approximate weight fraction in fly ash. The *D [4,3]* of fly ash was 41.51 μm. This type of fly ash contained particles mostly in the small size range <100 μm, 77%. As the plots shown in Figure 1a, the percentages of cenosphere recovery for the acetone/water ratio of 100:0 were significantly lower than those of the ratios of 20:80 or 0:100. This was also evident in the lesser fractions of particles with a density less than 0.79 g/cc contained in the fly ash sample collected in the specific density range of the 100:0 mixture. As a combination effect, such large fractions of small particles of the feeding fly ash are believed to make a significant contribution to the isolation–floating ability of the larger (to some extent meaning the lighter) particles in escaping from their agglomeration and floating to the upper surface [24].

An analysis of the particle size and weight percentage distribution of cenospheres is available in Table 2 and Figure 2. It is apparent that the dependence of the mean diameter on the medium proportions for the fly ash/medium ratios of 1:2.5 and 1:10 was approximately similar. The higher acetone contents, with a corresponding lower medium density, resulted in a larger size of cenospheres

being obtained. That is, for the acetone/water ratios of 100:0 and 80:20, the average particle size lay in the range of approximately 110 μm to 150 μm. For the ratios of 50:50 and 20:80, with a higher medium density, the average particle size lay in the lower range of 30 μm to 80 μm.

**Table 2.** Analysis of particle size and weight percentage size distribution of cenospheres collected from using different ratios of freshly prepared acetone–water mixtures, a 4-h soaking period, and fly ash/medium ratios of 1:2.5, 1:5, and 1:10. *D [4,3]*: diameter.

| Fly Ash/Medium Ratio | Acetone/Water Ratio | *D [4,3]* (μm) | Weight Percentage Size Distribution (%) | | | | |
|---|---|---|---|---|---|---|---|
| | | | <50 μm | 50–100 μm | 100–150 μm | 150–200 μm | >200 μm |
| 1:2.5 | 100:0 | 125.09 ± 2.71 | 18.46 | 24.16 | 23.69 | 16.23 | 17.46 |
| | 80:20 | 131.42 ± 2.79 | 18.50 | 21.19 | 23.49 | 16.63 | 20.18 |
| | 50:50 | 51.65 ± 0.67 | 63.09 | 23.02 | 7.66 | 3.24 | 2.99 |
| | 20:80 | 38.17 ± 1.21 | 68.98 | 25.74 | 4.98 | 0.30 | n/a |
| 1: 5 | 100:0 | 91.56 ± 2.99 | 31.85 | 31.22 | 19.17 | 9.98 | 7.79 |
| | 80:20 | 141.21 ± 3.01 | 11.48 | 22.89 | 25.66 | 18.2 | 21.77 |
| | 50:50 | 113.55 ± 1.10 | 18.43 | 30.33 | 24.77 | 13.81 | 12.66 |
| | 20:80 | 96.07 ± 1.33 | 29.80 | 30.96 | 19.62 | 10.01 | 9.61 |
| 1:10 | 100:0 | 147.76 ± 2.33 | 13.60 | 19.58 | 23.40 | 17.84 | 25.58 |
| | 80:20 | 110.57 ± 4.75 | 29.67 | 24.07 | 18.18 | 12.50 | 15.58 |
| | 50:50 | 60.67 ± 2.64 | 60.35 | 19.15 | 9.35 | 5.03 | 6.12 |
| | 20:80 | 82.85 ± 1.86 | 41.18 | 27.56 | 15.24 | 8.11 | 7.90 |

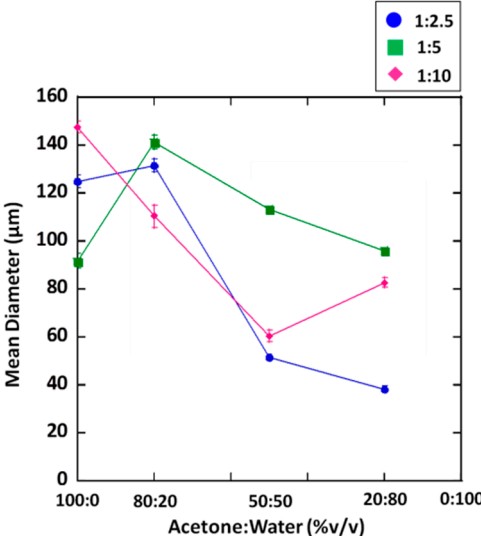

**Figure 2.** Mean diameters of cenospheres collected from using different ratios of freshly prepared acetone–water mixtures, a 4-h soaking period, and fly ash/medium ratios of 1:2.5, 1:5, and 1:10.

Two exceptions to this trend were observed for the fly ash/medium ratios of 1:5 and 1:10: the decrease in size with the increase in water content appeared to be inconspicuous. For the ratio of 1:5, the average particle size of the cenosphere fraction collected from using the acetone/water ratio of 80:20 (*D [4,3]*, 141.21 μm) was greater than that of the 100:0 ratio (*D [4,3]*, 91.56 μm), which was a mean size smaller than expected in the <100 μm region. Meanwhile, in the ash content ratio of 1:10, the average particle size of the cenosphere fraction collected from using the acetone/water ratio of 20:80 (*D [4,3]*, 82.85 μm) was larger than that of the 50:50 ratio (*D [4,3]*, 60.67 μm). This could be attributed to the disproportionate loss of the particles when collecting the cenosphere fractions. For instance, in the case of the ash/medium ratio of 1:10, the loss of fine particles could possibly be due to the pronounced effect of crystal formation on the surface of the fine ash particles and possible agglomeration creating large-sized aggregates; as a result, the particle size distribution is seen shifted to the larger size region.

The particle size distribution profiles of the collected cenosphere fractions are demonstrated in Figure 3 as a function of the fly ash content in the medium. Figure 3a,b reveals the asymmetric particle size distribution, and Figure 3c reveals the nearly normal Gaussian particle size distribution. The profile close to the normal Gaussian particle size distribution is regarded as the best one, considering that the population lies below $D_{10}$, $D_{50}$, and $D_{90}$ [23]. For certain applications, the uniformity of particle size and weight distributions of cenospheres is preferred for the use of cenospheres as fillers and additives, which affects the mechanical properties of the final products [32,39].

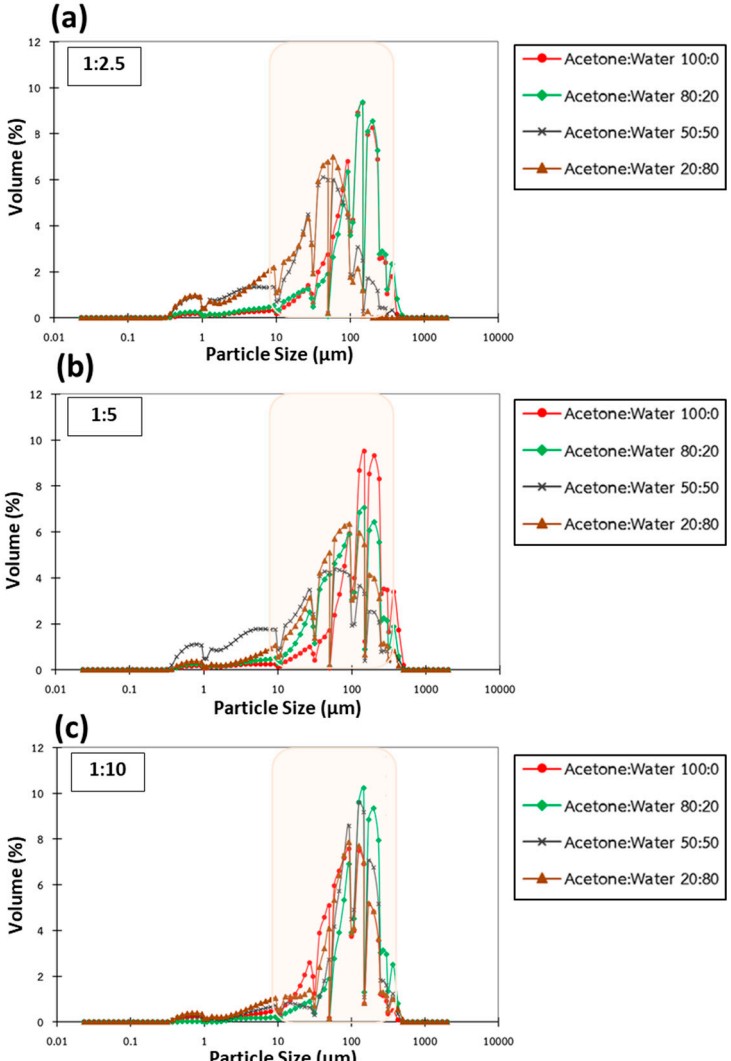

**Figure 3.** Particle size distribution profiles of cenospheres collected from using different ratios of freshly prepared acetone–water mixtures, a 4-h soaking period, and fly ash/medium ratios of (**a**) 1:2.5, (**b**) 1:5, and (**c**) 1:10.

At a 1:2.5 ratio, high fly ash concentration appeared to have a strong effect on the selective size distribution as a function of the medium fraction relating to density. The clear discrete groups of distribution for the large-sized fractions (100:0 and 80:20 ratios) and the smaller-sized fractions (50:0 and 20:80 ratios) are seen in Figure 3a. For the more diluted fly ash/medium ratio of 1:10, all fractions of the collected cenospheres appeared to have a similar profile of particle size distribution in the large-sized range. The dilution parameter in terms of low fly ash content in the medium facilitated the large-sized particles in the low liquid density to easily escape from the ash agglomeration and float to the surface with respect to the small-sized particles.

### 3.3. Chemical Composition

Table 3 shows the oxide compositions in fly ash and the collected cenospheres characterized by the XRF technique. The sum of $SiO_2$, $Al_2O_3$, and $Fe_2O_3$ in fly ash was 60.49%, which was in the range of 50%–70%, and the CaO content was higher than 5.0%, hence confirming its classification as class C fly ash (regarding ASTM C618) [40]. Contents of $SiO_2$ + $Al_2O_3$ + $Fe_2O_3$ > 50% in high-Ca class C fly ash exhibit cementitious properties that can be used as Portland cement replacement [41]. It should be noted that the chemical composition of cement involves both major and minor oxides, such as fly ash, except for the predominant content of CaO. The presence of high CaO in conjunction with $SiO_2$, $Al_2O_3$, and $Fe_2O_3$, forming calcium aluminosilicates and aluminoferrite hydrate, leads to the hardening of Portland cement during cement hydration. The chemical composition of cement in comparison to fly ash is available in the literature [42,43]. Fly ash recovered from the sink part was found to contain those three major components at 61.19%, with other compositions found to be insignificantly different from the fresh fly ash. CaO was not only embedded in the glassy spheres, but was present as free lime in fly ash. The composition of CaO in fly ash usually increases with decreased particle size [44] and has been found to be up to 15–40 wt.% for fly ash produced from lignite coal [45].

The cenospheres have a similar composition to fly ash, but are larger in size. The chemical compositions in the cenospheres were found to mainly consist of the macrocomponents of $SiO_2$, $Al_2O_3$, and $Fe_2O_3$. The $SiO_2$ content lay in the range from 36 wt.% to 56 wt.%, the $Al_2O_3$ content varied from 16 wt.% to 25 wt.%, and the $Fe_2O_3$ content varied in the narrow range of approximately 4 wt.% to 13 wt.%, accommodating the type of cenosphere separated from lignite coal fly ash from the Mae Moh thermal power plant, Thailand, in the magnetic category [46,47]. The $Fe_2O_3$ content in cenospheres of about 7 wt.% involved a magnetic phase based on defect magnetite, while the low $Fe_2O_3$ content of 2.5–3.5 wt.% was classified as nonmagnetic cenospheres [31]. It is worth noting that the presence of iron oxide in the aluminosilicate shell of cenospheres in terms of ferrosilite and relatively heavier elements such as Ca, Fe, and Ti, accompanied by a thick shell, could possibly make the cenospheres of density heavier than water, at 1.282 g/cc [24]. As seen in Table 3, for the ratios of 50:50 and 20:80 of acetone/water, the $Fe_2O_3$ content in cenospheres tended to increase with the increased water contents. Accordingly, those containing heavy elements in the cenospheres shell enable the cenosphere fractions to be collected in the relatively higher density of the medium (0.90–0.96 g/cc), as the data show in Table 1.

The cenospheres are a mixture of aluminosilicate mixed with a moderate amount of calcium oxide and ferric oxide and a slight content of S, K, Ti, and trace elements of Ti, Mn, Cr, and Cu. The presence of trace elements demonstrated their amounts in a limited concentration. Variation in the medium properties did not show a significant impact on the reduction of their concentration with a decrease in particle size.

**Table 3.** Chemical compositions of fly ash, recovered fly ash, and cenospheres collected from using different ratios of freshly prepared acetone–water mixtures, a 4-h soaking period, and fly ash/medium ratios of 1:2.5, 1:5, and 1:10 (nd is not detectable by the instrument). All data points were obtained from three measurements.

| Compound wt % | Fly Ash | Recovered Fly Ash | Acetone/Water (% *v/v*) | | | | | | | | | | | |
| --- | --- | --- | --- | --- | --- | --- | --- | --- | --- | --- | --- | --- | --- | --- |
| | | | 100:0 | | | 80:20 | | | 50:50 | | | 20:80 | | |
| | | | 1:2.5 | 1:5 | 1:10 | 1:2.5 | 1:5 | 1:10 | 1:2.5 | 1:5 | 1:10 | 1:2.5 | 1:5 | 1:10 |
| $SiO_2$ | 32.12 ± 0.29 | 30.90 ± 0.16 | 56.91 ± 0.55 | 51.20 ± 0.26 | 51.68 ± 0.22 | 50.80 ± 0.52 | 51.63 ± 1.73 | 50.57 ± 0.84 | 43.70 ± 0.73 | 45.02 ± 1.91 | 36.60 ± 0.43 | 37.95 ± 0.20 | 44.88 ± 1.23 | 46.03 ± 0.55 |
| $Al_2O_3$ | 13.82 ± 0.19 | 14.08 ± 0.24 | 25.16 ± 0.13 | 23.22 ± 0.16 | 23.02 ± 0.22 | 22.72 ± 0.61 | 23.29 ± 0.29 | 23.08 ± 0.33 | 19.87 ± 0.30 | 20.55 ± 0.77 | 16.34 ± 0.25 | 17.11 ± 0.14 | 20.3 ± 0.37 | 21.34 ± 0.09 |
| $Fe_2O_3$ | 14.55 ± 0.27 | 16.21 ± 0.45 | 4.65 ± 0.46 | 8.29 ± 0.17 | 7.55 ± 0.60 | 5.34 ± 0.92 | 8.37 ± 1.12 | 8.74 ± 0.38 | 11.62 ± 0.10 | 10.19 ± 0.15 | 13.19 ± 0.28 | 12.16 ± 0.18 | 9.08 ± 0.88 | 10.04 ± 0.31 |
| CaO | 24.49 ± 0.05 | 25.33 ± 0.63 | 8.10 ± 0.20 | 9.50 ± 0.04 | 10.39 ± 0.37 | 11.90 ± 0.95 | 10.2 ± 0.08 | 8.46 ± 3.81 | 16.79 ± 0.65 | 15.48 ± 1.08 | 22.93 ± 0.46 | 21.34 ± 0.34 | 16.35 ± 0.71 | 13.63 ± 0.45 |
| $SO_3$ | 12.03 ± 0.26 | 10.03 ± 0.27 | 0.81 ± 0.10 | 0.96 ± 0.05 | 1.27 ± 0.03 | 4.77 ± 1.55 | 1.61 ± 0.8 | 1.19 ± 0.05 | 3.46 ± 0.31 | 4.07 ± 1.65 | 6.76 ± 0.22 | 7.12 ± 0.11 | 4.35 ± 0.45 | 2.76 ± 0.17 |
| $K_2O$ | 2.31 ± 0.02 | 2.56 ± 0.03 | 3.74 ± 0.03 | 3.66 ± 0.03 | 3.69 ± 0.07 | 3.88 ± 0.03 | 4.05 ± 0.03 | 3.92 ± 0.16 | 3.62 ± 0.06 | 3.81 ± 0.02 | 3.27 ± 0.03 | 3.46 ± 0.03 | 4.1 ± 0.36 | 3.97 ± 0.08 |
| $TiO_2$ | 0.55 ± 0.02 | 0.61 ± 0.01 | 0.63 ± 0.02 | 0.77 ± 0.01 | 0.69 ± 0.04 | 0.59 ± 0.02 | 0.76 ± 0.03 | 4.24 ± 6.02 | 0.76 ± 0.00 | 0.75 ± 0.03 | 0.71 ± 0.01 | 0.67 ± 0.01 | 0.78 ± 0.04 | 0.75 ± 0.02 |
| MnO | 0.55 ± 0.01 | 0.14 ± 0.01 | nd | nd | 0.05 ± 0.00 | nd | nd | 0.06 ± 0.00 | 0.07 ± 0.00 | nd | 0.09 ± 0.00 | 0.10 ± 0.00 | nd | 0.07 ± 0.00 |
| $Cr_2O_3$ | nd | 0.55 ± 0.01 | nd | nd | Nd | nd | nd | 0.41 ± 0.00 | 0.06 ± 0.00 | nd | 0.06 ± 0.01 | 0.06 ± 0.00 | nd | 0.05 ± 0.00 |
| CuO | nd | nd | nd | nd | 0.03 ± 0.00 | nd | nd | 0.04 ± 0.00 | 0.04 ± 0.00 | nd | 0.04 ± 0.00 | 0.04 ± 0.00 | nd | 0.04 ± 0.00 |
| MgO | nd | nd | nd | nd | 2.45 ± 0.10 | nd | nd | 2.20 ± 0.00 | nd | nd | nd | nd | nd | 1.98 ± 0.05 |

Specifically focusing on the oxide iron–aluminosilicate ($Fe_2O_3$–$SiO_2$–$Al_2O_3$) system, a sum of $Fe_2O_3$, $SiO_2$, and $Al_2O_3$ higher than 80% is regarded as a cenosphere with good quality [23,48]. As seen in Figure 4a, the sum of the $SiO_2$ + $Al_2O_3$ + $Fe_2O_3$ enrichment for the 50:50, 80:20, 100:0 ratios was higher than 80 wt.%. This monotonically varying composition of cenospheres affected by the fly ash/medium variables for the reused conditions seemed to be fairly similar, except for the 50% acetone condition prone to the comparatively lower values. This result implies that the quality of cenospheres drastically reduced with more incorporation of water into the medium. A wide range of these compositions has been reported for cenospheres obtained from fly ash from different sources [27,32,33].

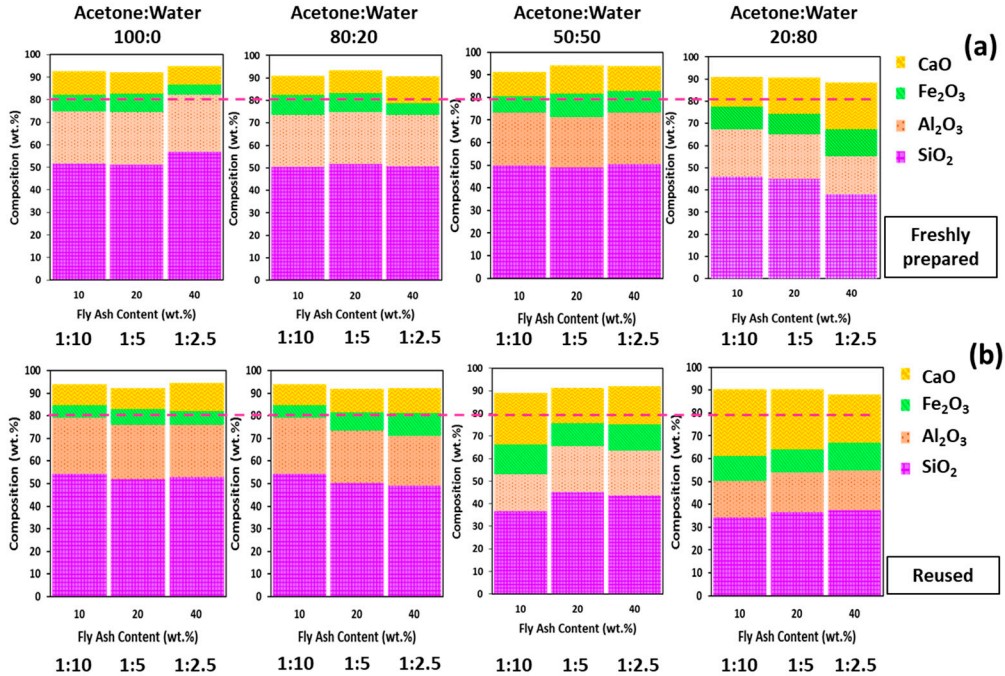

**Figure 4.** Stacked column plots of chemical compositions of cenospheres collected from using different acetone/water ratios, a 4-h soaking period, and fly ash/medium ratios of 1:2.5, 1:5, and 1:10. Data of (**a**) the freshly prepared and (**b**) the reused mixtures are presented for comparison.

For both the freshly prepared and the reused mixtures, an overview of the trend of the $SiO_2$ + $Al_2O_3$ + $Fe_2O_3$ composition seemed to show no certain correlation between the weight composition of these three components and the ash concentration in the medium. However, in the relation of size and composition, for larger particle size the $SiO_2$-$Al_2O_3$ content in the cenospheres is higher, as $SiO_2$ and $Al_2O_3$ are the main characteristics determining mechanical properties and inertness [32]. Particularly comparing the 80:20 to the 100:0 proportions, the increase in $SiO_2$ and $Al_2O_3$ contents with the increased acetone content appeared to be fairly inconspicuous. Thus, one could understand that this result suggests that the reduced density for the large-sized cenospheres was probably because of the hollow structure rather than the inherent chemical composition characteristics.

For the 50% and 80% water contents of the reused conditions, the higher contribution of CaO was seen more clearly. As seen from Figure 4b, the $SiO_2$ + $Al_2O_3$ + $Fe_2O_3$ composition in the 50:50 and 20:80 ratios apparently dropped due to the enhancement of CaO in the cenospheres: the occupied areas of CaO were obviously intense in the whole fraction of each stacked column. Summarizing this observation, such a rich CaO phase formation existed in tandem with a reduction in the particle size of cenospheres.

The CaO and $SO_3$ contents increased monotonically with an increase in the water content (see Figure 5a,b). The data for the freshly prepared and the reused conditions were demonstrated for comparison. The CaO content varied from 10 wt.% to 30 wt.%, and the $SO_3$ content varied from 1 wt.%

to 8 wt.%. The substantial rise of the weight percentage of CaO and $SO_3$ was clearly seen at 80% *v/v* water content in the mixture, in which the fractions of smaller particles were produced. Using 100% water as a medium and a 1:5 fly ash/medium ratio, the CaO composition found in the cenospheres was up to 36.21 wt.%.

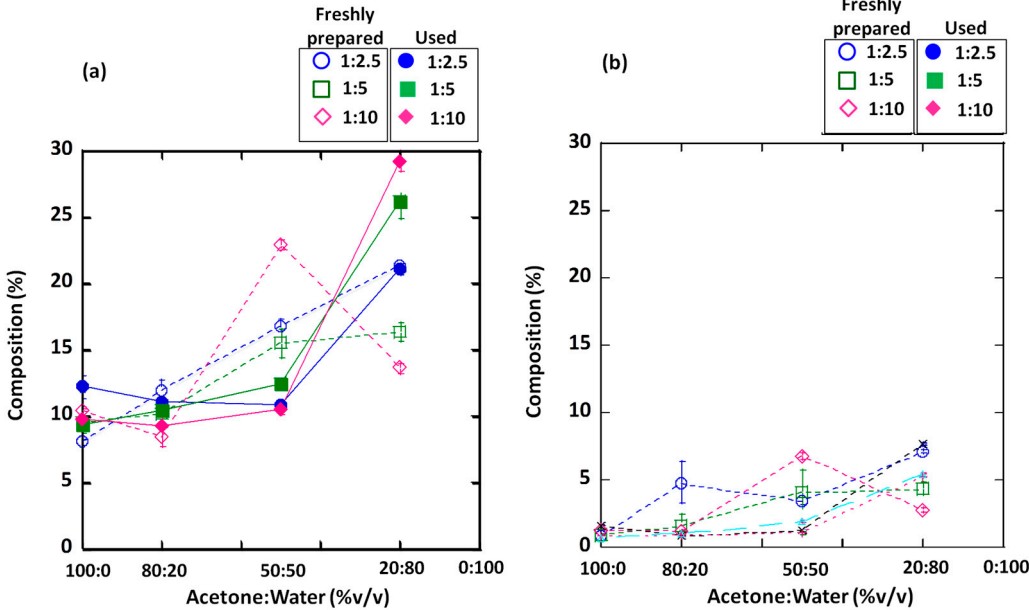

**Figure 5.** Composition of (**a**) CaO and (**b**) $SO_3$ in cenospheres collected from fly ash, using different acetone/water ratios, a 4-h soaking period, and fly ash/medium ratios of 1:2.5, 1:5, and 1:10. Data of the freshly prepared and the reused mixtures are presented for comparison.

Figure 6a,b shows the pair dependence of $SiO_2$–$Al_2O_3$ and $Al_2O_3$–$Fe_2O_3$ of the cenospheres, respectively. In Figure 6a, the $SiO_2$ content was dependent upon the $Al_2O_3$ content described in the linear equation. The trend revealed the same characteristic, that the dependence of $SiO_2$–$Al_2O_3$ was independent of the ash content in the medium. The $Al_2O_3$–$Fe_2O_3$ pair dependence in Figure 6b is described by a linear regression equation. This trend exhibited an increase in the $Fe_2O_3$ content with a decrease in the $Al_2O_3$ content. The results obtained from this analysis were in satisfactory agreement with the data ranges reported for the category of magnetic cenospheres [31]. $SiO_2$, $Al_2O_3$, and $Fe_2O_3$ participate in the composition of the glass phase and can differently affect the morphology and thus the properties of the aluminosilicate matrix of the cenosphere shell. Oxides of Si and Al involve the formation of mullite, and oxides of Fe and Al participate in the formation of ferrospinel [31].

As seen in Figure 6c, the cenosphere fractions obtained in this study revealed an interesting result of the constant $SiO_2/Al_2O_3$ ratios of 2.23 (± 0.027), 2.20 (± 0.014), and 2.21 (± 0.042) for the fly ash/medium ratios of 1:2.5, 1:5, and 1:10, respectively. This observation indicates a geochemical characteristic of the magnetic cenospheres that originated from two immiscible high-silica melts with a constant $SiO_2/Al_2O_3$ ratio forming such spheres [31]. The $SiO_2/Al_2O_3$ ratio is one important characteristic criterion of cenospheres in their applicability in a water-insoluble mineral-like framework. A $SiO_2/Al_2O_3$ ratio in the range of 1.2–3.5 is for different structures of feldspars [31]. A fraction of cenospheres with a $SiO_2/Al_2O_3$ ratio equal to 2.4 was used to produce microspherical zeolites of an NaP structure [49] and feldspar of cesium pollucite, a potential structure possessing great resistance to leaching in the long-term disposal of radionuclide applications [50]. A promising application profile is suggested upon this achievement, and this result can be used to initiate a feasibility study for the potential applications of cenospheres collected from high-calcium lignite fly ash.

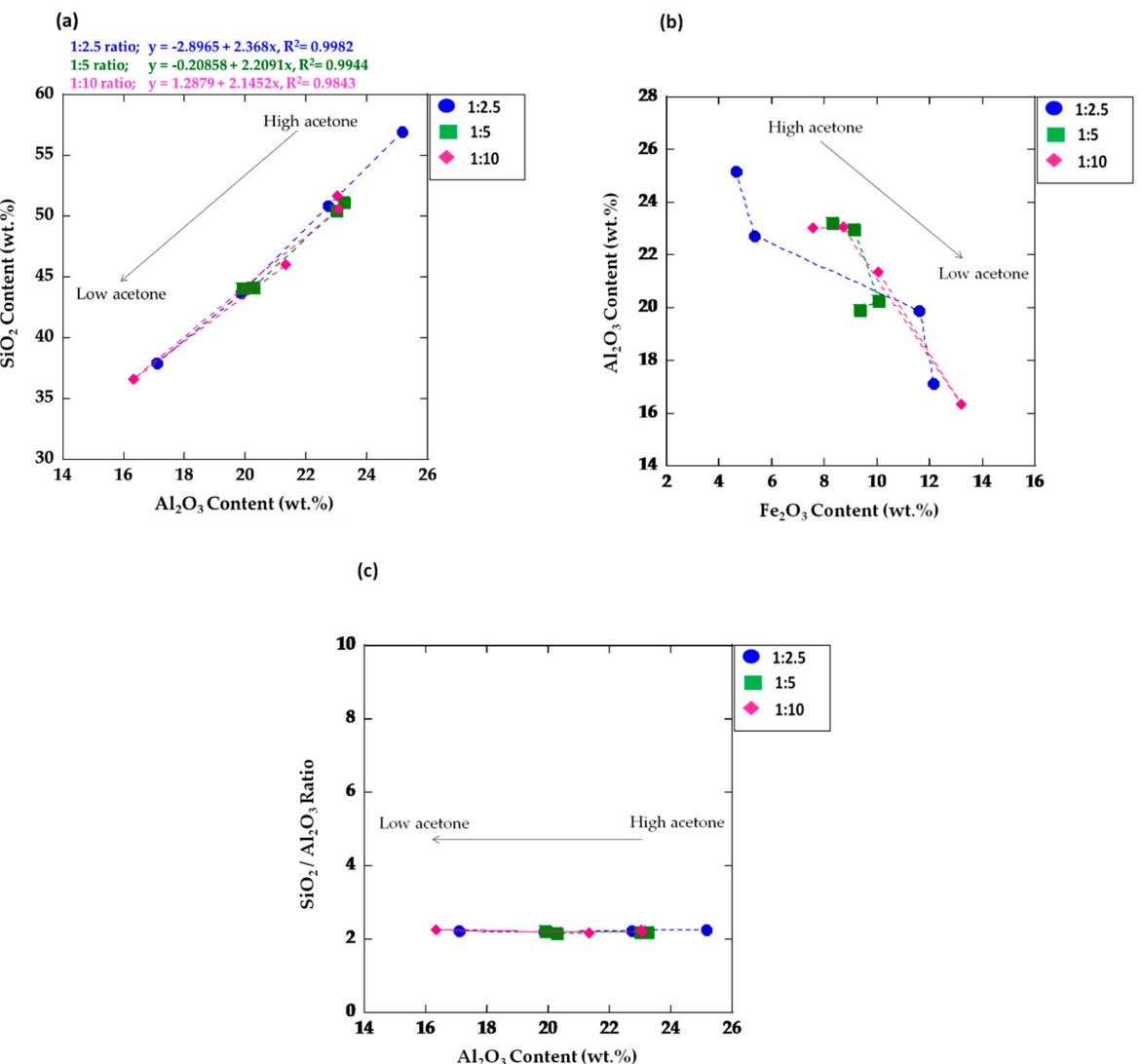

**Figure 6.** Dependence of (**a**) SiO$_2$ and Al$_2$O$_3$ contents, (**b**) Fe$_2$O$_3$ and Al$_2$O$_3$, and (**c**) the SiO$_2$/Al$_2$O$_3$ ratio and Al$_2$O$_3$ content. The data were taken from Table 3.

*3.4. Morphological Study*

The morphologies of cenospheres are presented in Figures 7–9. The as-received cenospheres were glassy, had open pores, and were spherical in shape, as seen in the XRF images in Figure 7. At a higher magnification, the cenospheres were found to exhibit a perforated shell and an irregular spherical shape of the particles (see Figure 8). It should be noted that the spherical shape of the particles is an important factor in improving, to some extent, flowability or workability in mixing cenospheres as fillers with a matrix such as polymer resin. The spherical shape, possessing a low surface area/volume ratio, requires a lower amount of binder matrix to wet the surface [24]. The cell wall thickness observed for the cenospheres collected from acetone–water mixtures varied from 2 to 30 μm. The broken cenospheres could be observed in a small proportion. The non-uniform wall thickness, with a thin wall and thick wall on the same particle, is seen in Figure 8a.

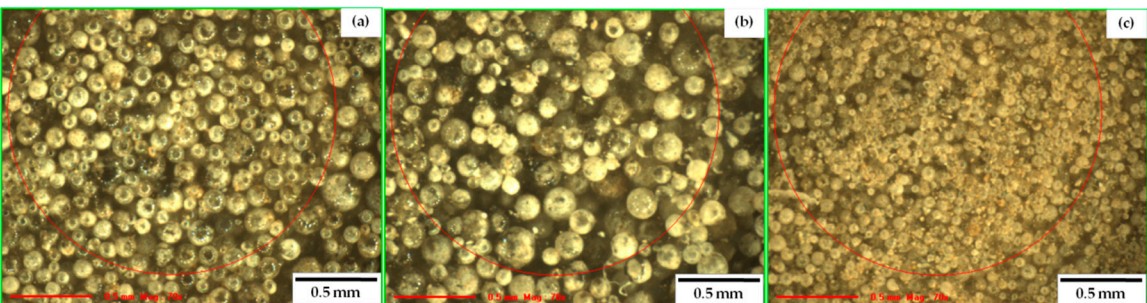

**Figure 7.** X-ray fluorescence (XRF) images of cenospheres collected from the freshly prepared mixtures at different acetone/water ratios of (**a**) 80:20, (**b**) 50:50, and (**c**) 20:80, using a fly ash/medium ratio of 1:5 and a 4-h soaking period.

Using higher acetone ratios of 80% and 100%, the surface of the larger cenospheres was found to be clean, no noticeable amount of small aggregates deposited on the cenosphere surface as seen for the samples collected from using high water content condition (see Figure 8g,h). The magnified surface of cenospheres, shown in Figure 9a, revealed various morphologies of the aggregates forming on the surface of cenospheres. Those precipitates are in various sizes and shapes such as spherical and nonspherical particles, needles, small spheres with whiskers, and corrugated surfaces. Such morphological characteristics of corrugated, rough surfaces, noticeably indicated in fine fractions, are classified as crystallite depositions on the surfaces of magnetic components such as iron oxides, which are mostly found on magnetic spheres [51–54]. In cenospheres with a high iron content, part of the outer surface contains heterogeneous regions with extended linear inclusions of ferrospinels, while on the inner surface, the pores contain the underlying aluminosilicate phase [31]. The formation of such aggregate particles on the surface may involve a nanoscale dissolution–precipitation mechanism for alkali activation reactions in an aluminosilicate phase. Fly ash and cenosphere reactivity is based on the alkaline dissolution of disordered aluminosilicate networks. Many studies have described the conceptual mechanism of an alkali attack on the ash surface, leading to sodium aluminosilicate hydrate N-A-S-H gel polymerization from inside and outside the particle shell and ultimately to reaction products precipitating on the inner and outer surfaces [55,56]. The hollow spheres were found to be broken, and hence it became possible to observe the inner wall surface (see Figure 9b).

With a prolonged soaking time up to 24 h, it was found that the entire surface of the cenospheres was covered with a remarkable amount of the aggregate crystals, as presented in Figure 9c,d. The elemental composition by EDX was further investigated for this sample, and the results indicated that those aggregates mainly contained $39.17 \pm 1.23$ wt.% Ca, $12.90 \pm 1.27$ wt.% C, and $47.93 \pm 0.04$ wt.% O: data were obtained from five measurements. It should be noted that the theoretical molar mass percentage ratio of calcium carbonate ($CaCO_3$) was 40.04 wt.% Ca, 12.00 wt.% C, and 47.96 wt.% O [57,58]. It could be presumed from this finding that the aggregates were prone to calcium carbonate according to such close relevancy. The large amount of aggregates covering the cenosphere particles was taken into consideration in the determination of the cenosphere recovery yield.

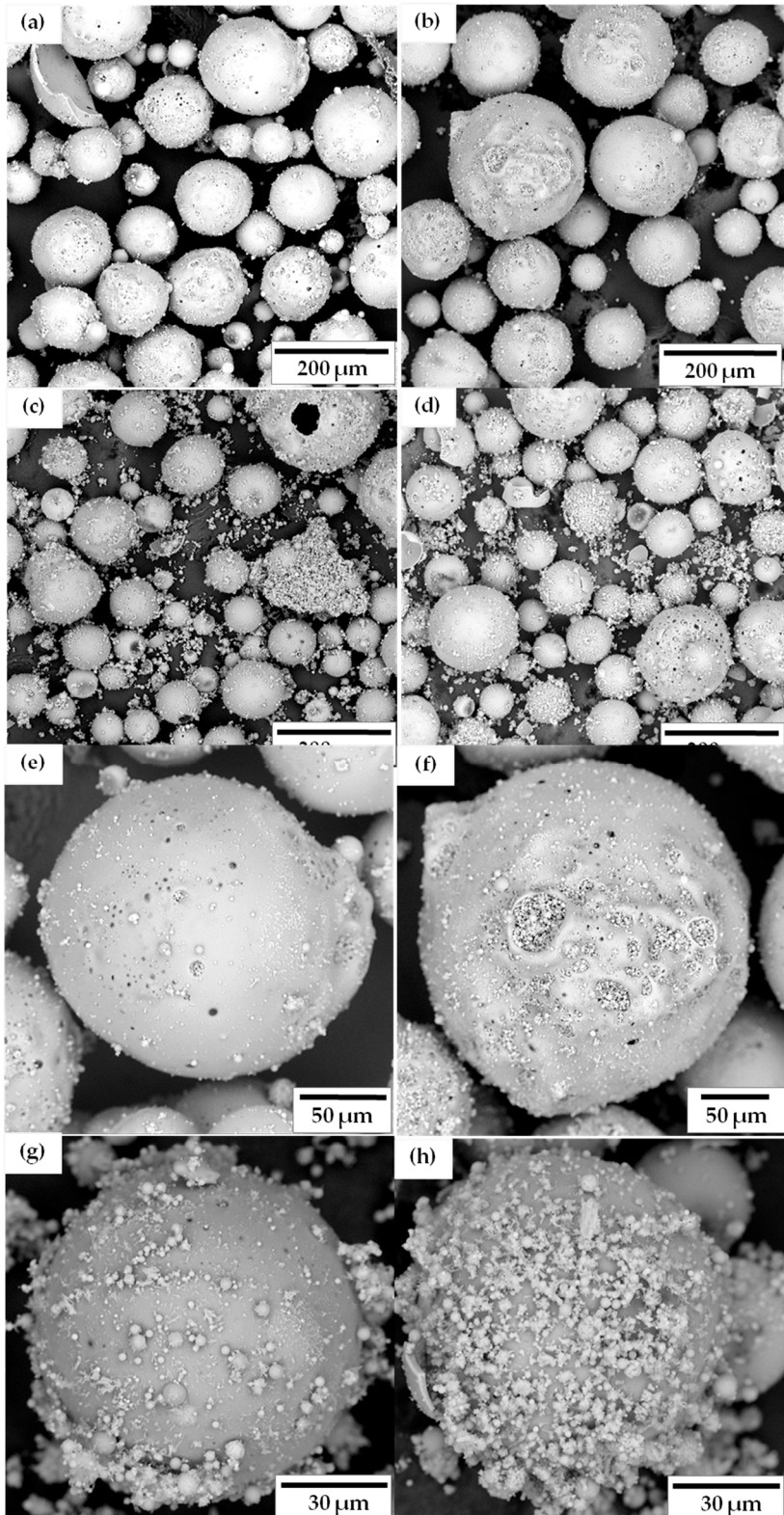

**Figure 8.** SEM images of cenospheres collected from the freshly prepared mixtures at different acetone/water ratios of (**a**,**e**) 100:0, (**b**,**f**) 80:20, (**c**,**g**) 50:50, and (**d**,**h**) 20:80, using fly ash/medium ratio of 1:5 and a 4-h soaking period. Here, (e–h) includes magnified SEM images selectively zoomed in on one cenosphere particle.

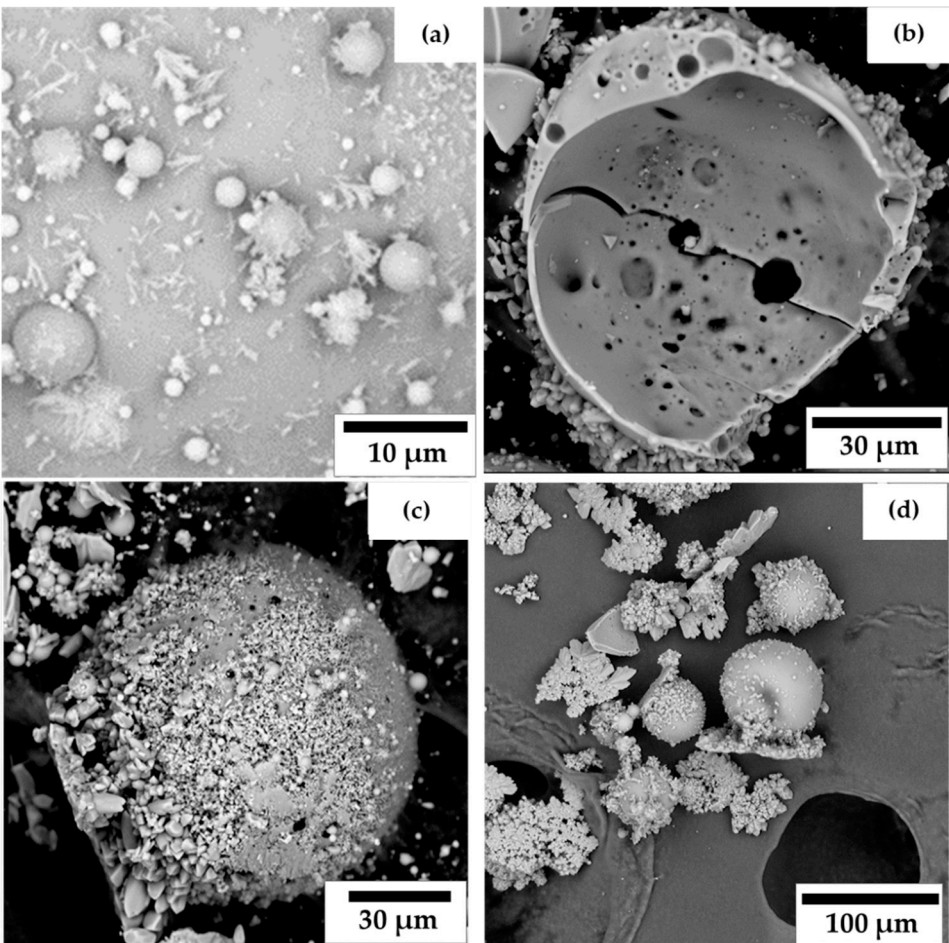

**Figure 9.** (**a**) The outer surface of cenospheres with different morphologies of aggregates deposited on the surface. (**b**) The broken particle of cenospheres, showing the inner wall surface. (**c**,**d**) Almost the entire surface of a cenosphere particle covered with a large amount of the aggregates: this particle was collected after a soaking period of 24 h using a 1:5 fly ash/medium ratio and 100% water as a medium.

*3.5. Carbonate Concentration*

The generation of calcium carbonate crystals on the ash surface due to the hydration of calcium silicate and free lime producing the calcium hydroxide product has been explained well in many reports [59,60]. The $Ca(OH)_2$ product can react with $CO_2$ gas and generate $CaCO_3$ crystals, as shown in Equation (2). The dissolution of calcium carbonate can also occur in water containing carbon dioxide:

$$Ca(OH)_2 \text{ (aq)} + CO_2 \text{ (g)} \rightarrow CaCO_3 \text{ (s)} + H_2O \text{ (l)}, \tag{2}$$

$$CO_2 \text{ (g)} + H_2O \text{ (l)} \leftrightarrow H_2CO_3 \text{ (aq)}, \tag{3}$$

$$CaCO_3 \text{ (s)} + H_2CO_3 \text{ (aq)} \leftrightarrow Ca(HCO_3)_2 \text{ (aq)}. \tag{4}$$

In this study, the ash dissolution raised the pH of the liquid medium substantially (to 13), along with silicate ion, $Al^{3+}$, and $Ca^{2+}$ concentrations. The original alkalis can recycle and play a key role in the subsequent alkaline activation reactions, finally forming secondary reaction products such as a carbonate phase [56].

It was of interest to investigate the influence of chemical component dissolution on the properties of the liquid medium. The $CaCO_3$ concentration in the liquid medium, after being used for cenosphere separation, was analyzed according to the standard measurement of APHA (2012), 2320 B, for carbonate alkalinity. Figure 10 shows the increase in the $CaCO_3$ concentration as a function of the acetone/water

ratio. Varying the water contents between 20%, 50%, 80%, and 100%, the attainable values were 5, 15, 38, and 40 mg/L, respectively. For 100% acetone content, less than 1 mg/L of the $CaCO_3$ concentration was detectable due to the threshold of detectability of the instrument, which is not shown in the graph. Incorporating more water, from 50% to 100%, a significant rise in the carbonate concentration was seen. This tendency was similar to the increase in the CaO composition (shown in the inset) found in the cenosphere fractions collected from these ratios.

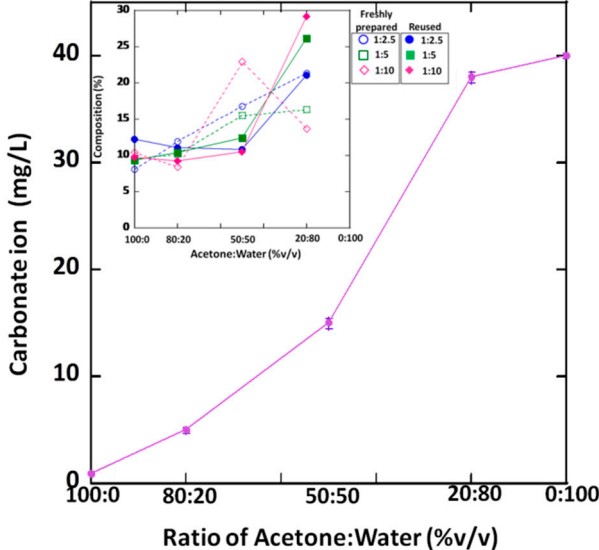

**Figure 10.** $CaCO_3$ concentration measured from the samples collected from the mixtures after being used for cenosphere separation using different acetone/water ratios and a fly ash/medium ratio of 1.5. The liquid samples were collected after a 4-h soaking period in the separation process.

Further considering the properties of the acetone–water mixture, the acetone content of the surface layer was remarkably higher than that of the bulk phase, and this difference became more pronounced at low acetone mole fraction in the bulk liquid phase [61]. Fábián et al. studied an interesting insight into the strong adsorption ability of the acetone molecules with solvents that was remarkably related to the presence of its apolar methyl ($-CH_3$) groups, leading to the unlike molecules tending to separate from each other, with the extent of the energy difference between the two molecules being in contact. The self-association ability of acetone and water was estimated to be relatively lower for the higher acetone fraction in the mixture system, and higher for the lower acetone fraction in the like molecule water. Thus, one could assume for this study that the diffusion of water-soluble species would be considerably influenced by this. Such a hindrance effect of the low self-association in the high acetone–low water mixture is believed to have played an important role in limiting the solubility, mobility, and reactivity of those water-soluble species and moving ions, thereby essentially having a strong impact on reaction kinetics and the formation of the reaction products, such as calcium carbonates, consequently affecting the separation efficiency, recovery yield, and quality of the cenospheres.

Additionally, given a clue for acetone–water interaction (considering a non-negligible factor in the floatability of particles), particularly in the high-acetone fraction but notwithstanding the more diluted fly ash content in the medium, the combination effect of the physicochemical properties of the acetone mixture may involve and probably has an increasingly important role in enhancing the miscibility of acetone and water molecular layers in the more diluted state, thereby leading to more interactions between the solvent molecules and the particles. The higher extent of this self-association was taken into account and reflects the fact that the high population of lighter particles by means of the

larger-sized particles floats to the surface, with the almost normal particle size distribution profile seen and described previously.

## 4. Conclusions

Due to a considerable concern about the occurrence of crystals on the ash surface during the wet separation process due to high calcium composition in class C fly ash, the research focus was accordingly on employing nonaqueous solvent and studying its fraction variation in crystalline inhibition. This work provided a systematic description on the effect of an acetone–water mixture on the recovery of cenospheres from high-calcium class C fly ash using the sink–float method. We presented an investigation of the physical properties, morphology, and chemical composition as a function of the acetone/water ratio. The influence of the fly ash/medium ratio was studied in comparison. Use of high water content was found to affect the separation efficiency, the recovery percentage, and the quality of the separated cenosphere product in terms of physicochemical properties. The relatively larger particle size of cenospheres with clean surface was obtained with higher acetone content, while the mixture ratio and the ash content feeding had to be manipulated properly to meet each specific desire. The proposed method is regarded as effective in achieving the size-selective distribution of cenospheres by tailoring the process parameters and the medium variables. The results for the freshly prepared and the reused mixtures were analyzed in comparison.

The relationships of physical properties-morphology-composition of cenospheres and the separation medium variables were described. The concentrations of the major chemical compositions revealed their dependence in a meaningful relation, clearly defining the cenosphere fractions collected from this study as magnetic cenospheres. The results confirmed the natural characteristic of how the cenospheres were formed through a constant $SiO_2/Al_2O_3$ ratio of about 2.2, complying with the conformation defined in the feldspars category. This work also presented an investigation of carbonate concentration in relation to the composition of CaO and the physicochemical properties of an acetone–water mixture. The self-association ability of acetone and water molecules is believed to play a predominant role in probably hindering the mobility and reactivity of ionic species in the mixtures.

**Author Contributions:** Conceptualization, S.Y.; methodology, S.Y.; formal analysis, S.Y. and P.T.; investigation, S.Y., T.I., and P.T.; T.I and P.T. performed the experiment and material characterization; P.T. partially analyzed the material properties; data curation, S.Y.; T.I. and P.T. participated in the data summary; writing—original draft preparation, S.Y.; writing—review and editing, S.Y.; visualization, S.Y.; supervision, S.Y.; project administration, S.Y.; funding acquisition, S.Y.

**Funding:** This research was funded by the Electricity Generating Authority of Thailand (EGAT), grant number 59-B104000-172-IO.SS03B3008261-MTEC, and the National Metal and Materials Technology Center, grant numbers P1651949 and P175126. The APC was funded by the National Metal and Materials Technology Center, National Science and Technology Development Agency, Thailand.

**Acknowledgments:** This research was mainly supported by the Electricity Generating Authority of Thailand (EGAT). The authors also acknowledge EGAT for providing fly ash for this study. We also thank the National Metal and Materials Technology Center, Thailand, for partial funding support and the instrument facilities. The authors also acknowledge Angkana Chumphu for her partial contribution in the analysis of the material properties.

**Conflicts of Interest:** The authors declare no conflicts of interest.

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
