# Peer review of "Separation of Cenospheres from Lignite Fly Ash Using Acetone–Water Mixture"

_applsci, doi:10.3390/app9183792_

Round 1
Reviewer 1 Report
Comments:
This paper proposed a method of separating Cenospheres from fly ashes. After addressing the following comments, the paper can be accepted for publication.
Review of existing method of separating Cenospheres and their advantages and disadvantages need to be highlighted? Compare the chemical composition of fly ash vs cement vs Cenospheres. Information of the chemical composition for fly ash and cement are available in “A detailed procedure of mix design for fly ash based geopolymer concrete” paper that should be cited while using this information. Where is the application of Cenospheres in construction sector? This hollow particle can be used as a filler to minimise crack propagation in concrete. Recently, the hollow particle has been introduced in polymer concrete for manufacturing composite beams [Ref: Flexural and shear behaviour of layered sandwich beams] and composite railway sleepers [Ref: Evaluation of an innovative composite railway sleeper for a narrow-gauge track under static load]. This information regarding application of hollow microsphere/ Cenospheres should be highlighted in introduction section. How effective the proposed method to separate Cenospheres in compare with the existing methods?Author Response
Reviewer 1
Comments and Suggestions for Authors
This paper proposed a method of separating Cenospheres from fly ashes. After addressing the following comments, the paper can be accepted for publication.
Review of existing method of separating Cenospheres and their advantages and disadvantages need to be highlighted?Answer Thank you very much for your suggestion. The comment has been well taken. The existing methods of cenospheres separation are classified into two groups: wet separation and dry separation. The review of these methods including their advantages and disadvantages has been added in Line 60-81.
Compare the chemical composition of fly ash vs cement vs Cenospheres.
Information of the chemical composition for fly ash and cement are available in “A detailed procedure of mix design for fly ash based geopolymer concrete” paper that should be cited while using this information.
Answer Thank you for your advice and kindly suggesting this paper. The discussion on the comparison of chemical composition of fly ash to cement and cenospheres has been added in Line 258-263, and a statement for cenospheres to be compared with fly ash has been addressed in Line 272.
Further, the suggested paper has already been cited as the Ref no. 44 (please see below), and all the references in the revised manuscript have been rearranged.
Reference:
[44] Ferdous, M.W.; Kayali, O.; Khennane, A. “A detailed procedure of mix design for fly ash based geopolymer concrete”, Proceeding of Fourth Asia-Pacific Conference on FRP in Structures (APFIS 2013), Melbourne, Australia, 11-13 December 2013.
Where is the application of Cenospheres in construction sector?
Answer The application of cenospheres in the construction sector has been added in the Introduction part, Line 40-45.
This hollow particle can be used as a filler to minimise crack propagation in concrete. Recently, the hollow particle has been introduced in polymer concrete for manufacturing composite beams [Ref: Flexural and shear behaviour of layered sandwich beams] and composite railway sleepers [Ref: Evaluation of an innovative composite railway sleeper for a narrow-gauge track under static load]. This information regarding application of hollow microsphere/ Cenospheres should be highlighted in introduction section.
Answer The information regarding the application of cenospheres in the polymer concrete and composite railway sleepers has been added in Line 42-44, with these two suggested references cited accordingly. [Ref no. 20-21] We thank the reviewer for giving us the suggestion to strengthen the part of construction application.
Reference:
[20] Ferdous, W.; Manalo, A.; Aravinthan, T.; Fam, A. “Flexural and shear behavior of layered sandwich beams”, Constr. Build. Mater. 2018, 173, 429-442. DOI. 10.1016/j.conbuildmat.2018.04.068
[21] Ferdous, W.; Manalo, A.; Van Erp, G.; Aravinthan, T.; Ghabraie, K. “Evaluation of an innovative composite railway sleeper for a narrow-gauge track under static load”, J. Compos. Constr. 2018, 22 (2), 04017050. DOI:101061/(asce)cc.1943-5614.0000833
How effective the proposed method to separate Cenospheres in compare with the existing methods?
Answer Thank you for the comment on this point. This work is the study of the wet separation, one of the existing methods aforementioned and added in Line 60-80, of cenospheres from the high calcium class C fly ash using the acetone-water mixture as medium. With a considerable concern on the occurrence of crystals on the ash surface during the wet separation process due to such high calcium composition in class C fly ash, the research focus was accordingly on employing non-aqueous solvent and studying its fraction variation on the crystalline inhibition. The high acetone content in the medium has shown its remarkable effect on the large particle size of cenospheres with clean surface to be obtained, whilst the concentration of the medium and the ash content feeding must be manipulated properly to meet each specific desire. The proposed method in this study is regarded as effective to achieving the selectively size distribution of cenospheres by tailoring the process parameters and the medium variables.
We have realized that this point has not been highlighted adequately and clearly in the previous version of manuscript. Thus, the statements regarding the reviewer’s comment have been added in the Conclusion part, Line 463-466, and Line 471-476.
More addition
We have also added an SEM image of Figure 9 (d) for a better completion of figure presentation.

Reviewer 2 Report
In this manuscript, the authors investigate the effect of acetone-water ratio (100:0, 80:20, 50:50, 20:80) and fly ash-to-medium ratio (1:2.5, 1:5, 1:10) on the separation of cenospheres from lignite fly ash. This manuscript is well written. This paper may be accepted in Applied Sciences with minor revision. The following points should be clarified:
1) Please explain why the density of the cenospheres fractions obtained from using the freshly prepared acetone-water mixture (100:0), and fly ash-to-medium ratios of 1:2.5 and 1:10 were not shown in Figure 1.
2) I would recommend the authors provide the standard errors of the average particle size of cenospheres in Table 2.
3) In-Line 186-187, the authors mentioned that:
“The higher acetone contents, with corresponding lower medium density, resulted in the larger size of cenospheres to be obtained.”
However, for the fly ash-to-medium ratio of 1:5, the average particle size of cenospheres collected from using the acetone-water ratio of 80:20 (D: 141.21 μm) is much larger than the average particle size of cenospheres collected from using the acetone-water ratio of 100:0 (D: 91.56 μm). In addition, for the fly ash-to-medium ratio of 1:10, the average particle size of cenospheres collected from using the acetone-water ratio of 20:80 (D: 82.85 μm) is larger than the average particle size of cenospheres collected from using the acetone-water ratio of 50:50 (D: 60.67 μm). Please explain why?
4) In Figure 6a, “R” should be changed into “R2”?
5) Please add some discussions about the reasons why the percentage of cenospheres recovery is so low for the acetone-water ratio of 100:0 when compared with recovery yield using the acetone-water ratio of 20:80 or 0:100?
Author Response
Reviewer 2
Comments and Suggestions for Authors
In this manuscript, the authors investigate the effect of acetone-water ratio (100:0, 80:20, 50:50, 20:80) and fly ash-to-medium ratio (1:2.5, 1:5, 1:10) on the separation of cenospheres from lignite fly ash. This manuscript is well written. This paper may be accepted in Applied Sciences with minor revision. The following points should be clarified:
1) Please explain why the density of the cenospheres fractions obtained from using the freshly prepared acetone-water mixture (100:0), and fly ash-to-medium ratios of 1:2.5 and 1:10 were not shown in Figure 1.
Answer Thank you very much for the comment on this point. We do apologize for the incomplete expression. We have made revision per the comment by revising the plots in Figure 1 (b), with the data for the density of the fly ash-to-medium ratios of 1:2.5 and 1:10 shown.
2) I would recommend the authors provide the standard errors of the average particle size of cenospheres in Table 2.
Answer Thank you very much for the concern to improve the quality of the data in Table 2. The standard deviations of the average particle size data have been added in Table 2, shown in the red color.
3) In-Line 186-187, the authors mentioned that:
“The higher acetone contents, with corresponding lower medium density, resulted in the larger size of cenospheres to be obtained.”
However, for the fly ash-to-medium ratio of 1:5, the average particle size of cenospheres collected from using the acetone-water ratio of 80:20 (D: 141.21 μm) is much larger than the average particle size of cenospheres collected from using the acetone-water ratio of 100:0 (D: 91.56 μm). In addition, for the fly ash-to-medium ratio of 1:10, the average particle size of cenospheres collected from using the acetone-water ratio of 20:80 (D: 82.85 μm) is larger than the average particle size of cenospheres collected from using the acetone-water ratio of 50:50 (D: 60.67 μm). Please explain why?
Answer Thank you very much for this observation and the comment. We realized that the previous discussion was unclear. Thus, we have revised the discussion on this part per the reviewer’s comments; please see the amended text in Line 208-225.
4) In Figure 6a, “R” should be changed into “R2”?
Answer Thank you for your suggestion. The “R” previously shown in Figure 6 (a) has been changed into “R2”.
5) Please add some discussions about the reasons why the percentage of cenospheres recovery is so low for the acetone-water ratio of 100:0 when compared with recovery yield using the acetone-water ratio of 20:80 or 0:100?
Answer Thank you very much for the comment. The discussion on why the percentage of cenospheres recovery is so low for the acetone-water ratio of 100:0 when compared with recovery yield using the acetone-water ratio of 20:80 or 0:100 has been added in Line 200-206.
More addition
We have also added an SEM image of Figure 9 (d) for a better completion of figure presentation.
